# What Is Rural Adversity, How Does It Affect Wellbeing and What Are the Implications for Action?

**DOI:** 10.3390/ijerph17197205

**Published:** 2020-10-01

**Authors:** Joanne Lawrence-Bourne, Hazel Dalton, David Perkins, Jane Farmer, Georgina Luscombe, Nelly Oelke, Nasser Bagheri

**Affiliations:** 1Centre for Rural and Remote Mental Health, University of Newcastle, Orange, NSW 2800, Australia; Joanne.LawrenceBourne@newcastle.edu.au (J.L.-B.); hazel.dalton@newcastle.edu.au (H.D.); 2Social Innovation Research Institute, Swinburne University of Technology, Hawthorn, VIC 3122, Australia; jcfarmer@swin.edu.au; 3School of Rural Health, University of Sydney, Orange, NSW 2800, Australia; georgina.luscombe@sydney.edu.au; 4School of Nursing, Faculty of Health and Social Development, University of British Columbia, Kelowna, BC V1V 1V7, Canada; nelly.oelke@ubc.ca; 5Centre for Mental Health Research, Australian National University, Acton, ACT 2601, Australia; nasser.bagheri@anu.edu.au

**Keywords:** rural adversity, rural mental health, rural communities, community wellbeing, social determinants, rurality, intersectionality, rural theory

## Abstract

A growing body of literature recognises the profound impact of adversity on mental health outcomes for people living in rural and remote areas. With the cumulative effects of persistent drought, record-breaking bushfires, limited access to quality health services, the COVID-19 pandemic and ongoing economic and social challenges, there is much to understand about the impact of adversity on mental health and wellbeing in rural populations. In this conceptual paper, we aim to review and adapt our existing understanding of rural adversity. We undertook a wide-ranging review of the literature, sought insights from multiple disciplines and critically developed our findings with an expert disciplinary group from across Australia. We propose that rural adversity be understood using a rural ecosystem lens to develop greater clarity around the dimensions and experiences of adversity, and to help identify the opportunities for interventions. We put forward a dynamic conceptual model of the impact of rural adversity on mental health and wellbeing, and close with a discussion of the implications for policy and practice. Whilst this paper has been written from an Australian perspective, it has implications for rural communities internationally.

## 1. Introduction

A growing body of literature recognises the profound impact of adversity on the mental health outcomes for people living in rural, regional and remote areas (referred to as ‘rural’ forthwith) [1,2]. Mental health is integral to our overall health and how we function in society [3]. While it is estimated that approximately 45% of all Australians aged 16 to 85 years will experience a mental illness in their lifetime [4], the high prevalence of psychological distress and untreated (or undiagnosed) mental illness [5] in rural and remote communities warrants specific attention.

Recognising that rural health outcomes are often poorer than those achieved in large cities [6], Bourke et al., 2012, produced a framework to understand the drivers of rural health outcomes in Australia in different rural contexts [7]. This six-factor framework highlights that, when compared with urban settings, rural places are impacted by geographic locality, ‘rural locale’ (the setting wherein social relations are formed), local health responses, broader health systems, social structures and finally the power relations between different levels of government and community members. These structural and cultural factors start to provide some explanation as to why rural mental health outcomes have proved to be enduringly problematic and resistant to attempts at reparation by policy makers and service providers.

Globally, the United Nations has highlighted vulnerable populations in rural and remote regions that do not have the same access to satisfactory health and wellbeing resources, although they have the same rights to do so [6]. This speaks to inequities and social justice issues. The cumulative impact of environmental, economic and social adversities is of concern and has become a public health priority [7], with rural and remote Australia no exception.

The recent Orange Declaration [8] addressed rural mental health issues and service challenges in Australia, concluding that current incremental approaches to improvement are unlikely to be effective and that integrated systemic solutions are needed. The paper highlighted the distinct features that people living in rural and remote places face in times of adversity. It called for collaborative efforts to improve the mental health of rural residents through research, policy development, service design and delivery, and community members initiatives. Importantly, it emphasised contextual variance, that rural places differ from each other as they do from metropolitan settings, thus place-based understanding and responses show greater promise than one-size-fits-all approaches.

Reconciling policy that is built for the average person in an average place to the contextually rich and variable experience of an individual in a particular place [9] requires a sophisticated understanding of place and its people which must be data informed and evidence based if effective solutions are to be implemented successfully. To do this, we need to understand that mental health care systems are complex adaptive systems [10,11]. Such systems are best viewed through an ecosystem lens [12] in which informal supports are key parts of the system, not just the health professionals and formal services. These informal supports keep people well and support them during illness and into recovery. When this is acknowledged, it highlights the interconnection of carers, family and community in an individual’s mental health. The mental health ecosystems approach takes “a whole-systems approach to mental health care facilitating analysis of the complex environment and context of mental health systems, and translation of this knowledge into policy and practice” [12].

Looking to other disciplines has shed light on rural health, including that of place and the intersection of varying combinations of social, economic and cultural factors [13,14,15]. Indeed, Farmer et al. note that ‘place is the omnipresent, but often unremarked variable in rural health research’ [15].

In this paper, we have built on the rural adversity model put forward by Hart et al. nearly a decade ago [16], in light of developments in our understanding of rural health, rural mental health, disaster response and recovery and the contribution of interdisciplinary scholarship. This conceptual paper draws upon a review of the published literature, theory synthesis and adaptation [17,18] and critical discussions from an online workshop involving experts from across Australia to address the questions: what is rural adversity, how does it affect wellbeing and what are the implications for action?

## 2. Methods

Building upon our experience and the framework developed in the Orange Declaration on Mental Health [8], we determined to refine our understanding of rural adversity. Key to this was to adopt a place-based and population health view and to explore the evidence widely through a literature review and expert deliberation. Our methodology is outlined in Figure 1, with the brief, or provocation, serving to inform the development of a discussion paper and convening of an expert interdisciplinary group (Appendix A) to review the discussion paper and work towards a consensus understanding of rural adversity.

The discussion paper was guided by the brief and a broad-ranging initial search in the fields of rural adversity (including disaster); rural epidemiology; theories and frameworks of rurality; rural risk and protective factors; and the impact of rural adversity on mental health. PubMed and Scopus databases were searched for academic papers and grey literature was found via Google and Google Scholar searches, with a date range of 2000–2020. After title and abstract screening across the multiple searches outlined above, 268 papers were selected and read in full by author 1. Of these, 168 were included in the discussion paper with an additional 39 papers included from in-reference citation and 19 papers from separate searches to explore concepts more fully. In total, 193 papers were included in the discussion paper.

For the expert workshop, the discussion paper was circulated two weeks in advance. A summary presentation was given and the group examined the logic of the paper, identified gaps in thinking and evidence, and sought to improve the concept and understanding of rural adversity. Key themes were agreed at the workshop. Authors 1 and 3 took detailed notes and a workshop record, summary notes and key themes were elaborated and refined. This informed further development of these themes and the revision of the rural adversity model as proposed by the Centre for Rural and Remote Mental Health in 2011 [16]. The conceptual paper was drafted by author 1 with guidance from authors 2 and 3. Upon a full draft, the full interdisciplinary group were invited to feedback and/or contribute to the paper as a co-author. Then the model and conceptual paper were refined iteratively. This conceptual paper represents a theory adaptation approach [18], in which we have reviewed and revised our previous understanding and model of rural adversity [16].

## 3. Results

### 3.1. Workshop—Rethinking and Reframing Rural Adversity

A summary of the key themes identified in the workshop is outlined in Table 1. The discussion paper addressed five key areas: rural adversity including disasters, rural epidemiology, theories and frameworks of rurality, rural risk and protective factors, and the impact of rural adversity on mental health.

### 3.2. How can we Explore Rural Diversity?

Rural locale is more than location and place. Rural areas share overlapping similarities and display distinct differences. Environmental dimensions of geography, economy, goods and services, culture and society (people) intersect simultaneously and have a multi-directional relationship with public health, mental health and wellbeing [15]. Contextualising theories and concepts of ‘intersectionality’ [13,14,19,20] can contribute to a relational understanding of rural variability that can point to different levels of disadvantage in different rural communities.

Rural communities are strongly associated with agriculture, primary industries, wilderness and desert areas, oceans and rivers; the corollary of this is that many rural people rely (directly or indirectly) on the land for work, income, home and connection [21,22]. Having a connection with the land is known to promote mental health and wellbeing for many agricultural families and Indigenous peoples [23,24]. Indeed ‘topophilia’, a concept that describes the affective bond one holds to a place, is positively associated with quality of life [25].

Rural populations have greater exposure to changes associated with the land. The impact of changes in the environment and climate on public health and wellbeing is increasingly acknowledged in health policy [26]. It is argued by some that adversity impacting the environment, whether natural or man-made, can lead to a disconnection with the land [24,27,28], which can challenge mental health. ‘Solastalgia’ describes the emotional distress that occurs when the land is under threat, degraded or different [29], and has been used in policy responses on the impact of climate on public health and wellbeing [26]. The mechanism proposed is a loss of trust, faith and reliance on the land affecting the positive benefits of a rural lifestyle and aggravating the socioeconomic disadvantages of rurality.

Providing and accessing health and support services across large and diverse areas in rural Australia is a widely recognised and enduring problem for all stakeholders. Scope of practice and services then become necessarily generalist and specialised services are unlikely to be available locally. Staff turnover, funding instability and disruption of access due to adverse events may mean that rural services are precarious [30]. Living in rural areas that are in decline, can negatively impact the mental health of residents [31,32,33]. Disruptions to the local economy can flow from global commodity price fluctuations leading to rapid closure of businesses, loss of jobs and displacement in communities with few alternatives and disruption of social capital [34].

These losses may cause population movement to places of higher amenity [35,36]. The viability of smaller rural communities and the added pressure on limited access to and inequitable distribution of quality regional health services is of concern to the Australian Government [37]. Reductions in rural populations through urbanisation leads to a cycle of depletion in community resources, services and support systems in some rural areas and added pressure for others [21]. The mental health of those who are left behind is challenged by the disruption to communities, family separation and diminished support networks, loss of businesses, schools and health services. Furthermore, there is deskilling due to reduced employment opportunities or increased workloads due to fewer available staff. Thus, population migration contributes to vulnerability in rural communities, reducing the capacity to mitigate against future adverse events.

Intersectionality takes the social determinants of health further by looking at the intersections of these determinants, particularly looking at how systems of power such as racism, classism, heterosexism interconnect with individual categories of difference (e.g., ethnicity, sex, gender). The inclusion of social categories of difference is important if we are to understand the impacts of power, complexity and inequity on the experiences of individuals [20]. The distribution of power can determine health inequities even unintentionally by not including diversity. Rural communities face many inequities including lower socioeconomic status, lower levels of education, fewer healthcare services, and challenges in accessing services. Individuals who live in rural communities experience these differences (ethnicity, sex, gender, disability) differently than urban communities. An understanding of rural adversity must extend beyond the individual social determinants of health and focus on how intersections impact an individual’s experience of mental health and wellbeing and the associated power dynamics.

Geospatial analyses, e.g., [38,39,40] and the Integrated Atlases of Mental Health Care, e.g., [41,42,43,44,45] are highly important decision-making tools to visualise the pattern of rural diversity or adversity and support an integrated and systematic way of collecting information from multi-layered rural ecosystems (communities). Advanced geospatial analyses offer a fundamental capacity to quantify and visualise variations and interaction between contextual factors (i.e., built and social environments, geographic isolation and environmental risk factors, and limited services and resources) and their impacts on rural adversity. This generates evidenced-informed knowledge to design tailored interventions and plans to mitigate rural adversity and empower rural people. We can conclude that rural communities are both diverse and dynamic hence the importance of an ecosystems approach to conceptualising rural adversity more comprehensively. Multi-dimensional representations of rurality reinforce the importance of understanding the effect of context in rural mental health care [46].

### 3.3. What does Adversity Look Like from a Rural Perspective?

Adversity is commonly understood as a difficult situation or hardship. At some time, everyone faces loss of employment or income, disability, serious illness, bereavement or sudden changes in circumstances. These adverse events are usually borne by the individual, with rippling impacts out to family and community. However, the reverse may happen when global or wider social adverse events have a rippling effect through the community to the individual. Individual adverse life events are met with a sense of loss, different forms of grief and psychological distress, and link to concepts of endurance, uncertainty, suffering, and hope [47].

Figure 2 describes the varying onset and duration of different adversities over time.

Traditional approaches to adverse events describe phases associated with the pre-adverse event period (susceptibility) of baseline strengths, vulnerabilities and resilience (adaptability) potential (Figure 2, line A) which are likely to be important predictors of individual and community outcomes [48]. Rapid onset adverse events (Figure 2, line B) may have lasting consequences such as the loss of productive capital, homes, livestock or people. These events may be sequential—for example, in Australia, flood can follow drought, or simultaneously weeds or locusts accompany low rainfall. When long-term adverse events such as drought occur (Figure 2, line C), the phase edges are less clear, in terms of when it starts, its end point and with a long recovery. Long-term adverse events may be exacerbated by uncertainty about the duration or severity of potential loss and the risk associated with mitigation strategies such as replanting, restocking or taking on additional debt. Individual life-course adverse events such as bereavement, serious illness, relationship breakdown or financial hardship (Figure 2, line D) may contribute to personal or baseline vulnerability or overlay and exacerbate the impact of other adversities. Figure 2, line (E) maps the intensity of impact comparing rapid onset with slow onset adversities.

It is important to note that the experience of adverse events is not always discrete and independent, but rather can be experienced sequentially or contemporaneously and have a cumulative impact on individuals, families and communities, in terms of mental health and adaptive capacity [49,50]. Cumulative impacts may then impact on the response, recovery and adaptation phases with the experience of adversity on individuals and neighbouring communities may differ markedly.

Taking an ecosystem view of adversity, the individual experiences of adversity can, at times, be overshadowed by periods of larger community-based adversities such as drought, fire, and floods (see Figure 3). These events may be and are often understood as, natural disasters with phases of preparedness, response, recovery and mitigation [48] but in practice, these events are complex with an interplay of challenges. Universal adversity, of which the COVID-19 pandemic and global economic recession are clear examples, may overshadow and exacerbate community-level adversities. Individuals and communities may find that expected forms of support are limited or absent since attention and resources are focussed elsewhere [51]. Moreover, this is deeply challenging in rural communities since systemic inequities already exist.

Framing rural adversity within an ‘ecosystem’ approach recognises the complexity of rural communities in which people, economies, societies, cultures, and contextual factors such as geography, climate and infrastructure impact at various individual, community and global levels, and in different circumstances (Figure 3). Rural and remote living involves a complex interlinking of these different system components. No one component or factor captures the essence of rural adversity as many features are interdependent; if one component faces adversity, others will be impacted.

This ‘ecosystem’ perspective also implies an understanding of ‘intersectionality’ [19,20], as described above.

Within rural communities, there may be individuals and households which are particularly vulnerable to adversity due in part to previous adverse events or a combination of such events. At the community level, we may see varying levels of precariousness [53] as available social and economic capital proves insufficient to meet a combination or succession of adverse events. Rural communities are sensitive to change [53], hence our focus on the impact of adverse events which fall disproportionately on rural and remote residents.

While it is convenient to map adverse events linearly, they are both systemic and cyclical and we will discuss these issues below.

### 3.4. How does Rural Adversity Impact on Wellbeing and What Are the Opportunities for Interventions?

One of the key aspects of rural adversity as originally postulated is the potential for a spiralling cycle of adverse events impacting on the physical environment (natural and built), reducing social and economic capital, which may lead to poor mental health and wellbeing reducing the capacity to mitigate future or continuing adverse events [16]. The model implicitly highlights that within an ecosystem view that mental ill-health has multiple bio-psycho-social and environmental causes and single medical or biological solutions are unlikely to prove successful [12]. Thus, we contend that developing a better understanding of this process has significant implications for all stakeholders interested in improving rural mental health outcomes.

We propose a dynamic conceptual model (see Figure 4) to demonstrate how rural adversity may impact on mental health and wellbeing in rural and remote communities and where interventions may be needed. This revised model draws upon earlier research about the prolonged Millennium drought in rural NSW [16] and emphasises that further adversity may exacerbate the vulnerability of individuals and the precariousness of the community and its constituent business and institutions.

Our model starts with the recognition that adverse events comprise acute and discrete phenomena but viewed from the perspective of rural residents and communities, they must be understood in combination and in sequence. Our schematic presentation must allow for combinations which may vary considerably between rural communities that may appear similar in population or scale. Such combinations of adversities put pressure on the physical environment and on the human or built environment and loss of amenity through drought, fire or flood may be accompanied by loss of buildings, roads bridges and other critical infrastructure.

Such losses impact on individuals, households and communities and pose challenges for rebuilding, recovery and in some cases continued residence or employment in a particular community. The cumulative losses may undermine social and economic capital and hence community wellbeing. For instance, loss of businesses, loss of employment and loss of income such as tourism, reduce community wellbeing and social and material resources necessary for recovery or adaptation.

Reduction in community wellbeing is not spread equitably. Particular sectors may suffer disproportionately as public sector employment is protected from commercial pressure while contractors and unskilled employees lose work. This may limit the informal support available in the community and compound inequality and disadvantage. This combination of human and material loss and reductions in social and economic capital and social support will be associated, at the population level, with a range of mental health problems including increases in psychological distress, increased substance use and in some cases post-traumatic stress and self-harm behaviours. This cycle continues and the community’s ability to mitigate further adverse events may be compromised.

There are several points where interventions may be helpful and as a complex system it will be necessary to act at a number of points and perhaps over an extended period of time. Investments in the physical environment are common following climatic adverse events and involve government, charitable and private resources. The use of government funds, charitable collections, insurance payments, loans and grants all have implications which go beyond the scope of this paper.

Secondly, support for individuals and communities to address loss and grief includes personal and financial counselling, assistance to navigate services and are often offered by outsiders for a limited period of time and funded by government agencies. Such support may be unaware of or unconnected with, local resources such as primary care and other providers.

Thirdly, interventions to improve community economic and social wellbeing include those provided by a range of rural development agencies charged with investing in business and skills opportunities to boost business and employment and thereby increasing social wellbeing.

The fourth set of interventions include the community development activities designed to increase social capital through collective action to advance locally determined priorities. These are often relatively low-cost investments but may have considerable benefits in building local capability for collective action, governance and adaptation.

The fifth point for interventions includes the provision of services to promote mental health and treat mental illness. Since specialist services are often in short supply in rural communities this may imply the development of new service models, with appropriate supervision and quality assurance mechanisms, that maximise the contribution of primary care and utilize a variety of technologies.

Risk mitigation is the sixth point of intervention in our model and may require political support and significant investments. The recent Australian experience of carting water to drought-affected communities implies that attention is needed if secure water supplies are to be assured for rural communities.

These intervention categories are designed to demonstrate that a suite of interventions are needed if rural communities are to flourish and to avoid the threats to wellbeing and mental health posed by the cycles of events we have described as rural adversity. In a complex rural mental health ecology, the suite of interventions must have sufficient variety or range to address the range of challenges that occur at individual, household, community and broader systems levels [54].

## 4. Discussion

We have proposed a dynamic conceptual model of rural adversity that builds on and adds to the previous model proposed by Hart et al., 2011 [16]. The model assumes a rural mental health ecology perspective with an analysis of the types and impacts of life-course, rural and systemic adverse events and a range of interventions that may mitigate the impacts on mental health and wellbeing.

This model does not assume that rural adversity comprises discrete adverse events requiring short-term interventions but rather events that must be understood sequentially and cumulatively, within a complex rural ecology. We have noted that rural individuals and communities are subject to rural adverse events which occur alongside life-course adversities and stressors and systemic adversities such as the COVID-19 pandemic and the subsequent recession.

The concept of intersectionality highlights that individual and community experience is complex. Rural residents may be homeowners, firefighters or emergency service volunteers, business owners or employees, members of families and neighbours. Their experience of adverse events takes many forms and one loss may be exacerbated by another. Human and material losses may be complex and the risk of poor mental health outcomes is considerable and compounding. The expertise of rural residents and their central role in experiencing and addressing rural adversity should be recognised and incorporated in policy development and action. Approaches such as deliberative dialogue may help in this regard [55,56].

New methods for collecting, analysing and displaying evidence provide new opportunities for deeper understanding and more coherent interventions. The use of visual data and models may help explain rural health issues to a wider audience, e.g., [39,40,41,45,57]. Such data could enable the planning and evaluation of interventions using relevant and timely data. This could be particularly useful in identifying groups of individuals and communities that are at particular risk.

This paper has a number of limitations. The range of relevant disciplines is large and we could not do justice to insights from the full range of demographic, epidemiological, psychological and social sciences. The paper is not a systematic review nor is it a data-driven description of rural mental health and its deficiencies. The paper does not adequately address the perspective of Aboriginal and Torres Strait Islander peoples, nor vulnerable or marginalized populations such as culturally and linguistically diverse people, LGBTIQ people and refugees. The authors cross academic and clinical boundaries and are engaged in research and service provision. Most importantly they are engaged on a daily basis in attempting to support better responses to the distress caused by the combination of adverse events experienced disproportionately by rural and remote residents and to support rural service delivery in difficult environments.

Recognition of rural adversity as a complex determinant of mental health and wellbeing implies that integrated and collaborative approaches are needed to develop and implement combinations of interventions that are tailored to local needs, based on evidence and driven by data, with local leadership and community support wherever possible. In Australia, this might imply a new emphasis on integrated regional mental health and suicide prevention planning in defined populations alongside attempts to mitigate future adverse events and build community capital and resilience [58].

## 5. Conclusions

In conclusion, we believe that a model of rural adversity located in an ecological model of rural communities provides a better path to promoting mental health and wellbeing and mitigating the health effects of rural adversity. This implies changes to the disjointed responses to adverse events and is particularly important for the wellbeing of rural communities.

Australia is not the only country challenged in this way. From a mental health and wellbeing perspective, developing a global understanding of international rural communities, similarities and differences contributes to the overarching concept of rural adversity. This paper is written from an Australian perspective, and although each rural context has its differences within and between rural locales, we anticipate that this paper will resonate internationally.

## Figures and Tables

**Figure 1 ijerph-17-07205-f001:**
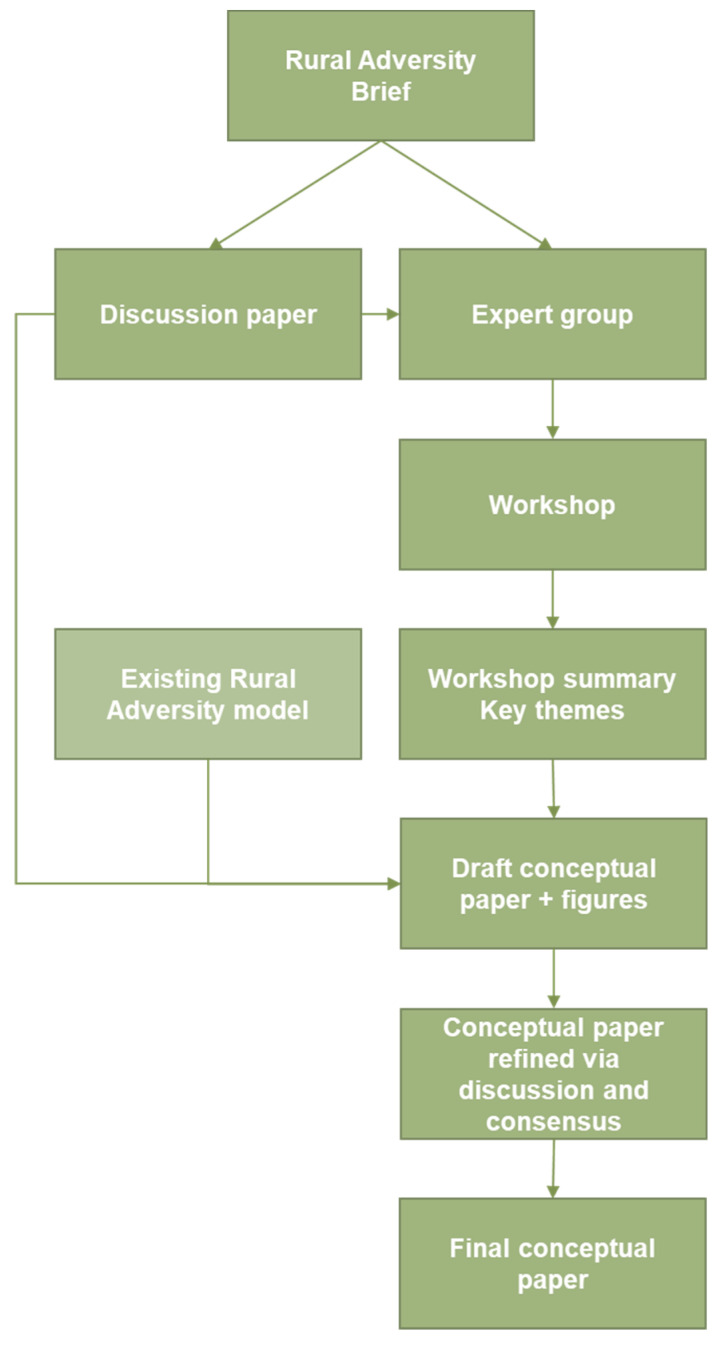
Schematic showing the staged development of the rural adversity conceptual paper.

**Figure 2 ijerph-17-07205-f002:**
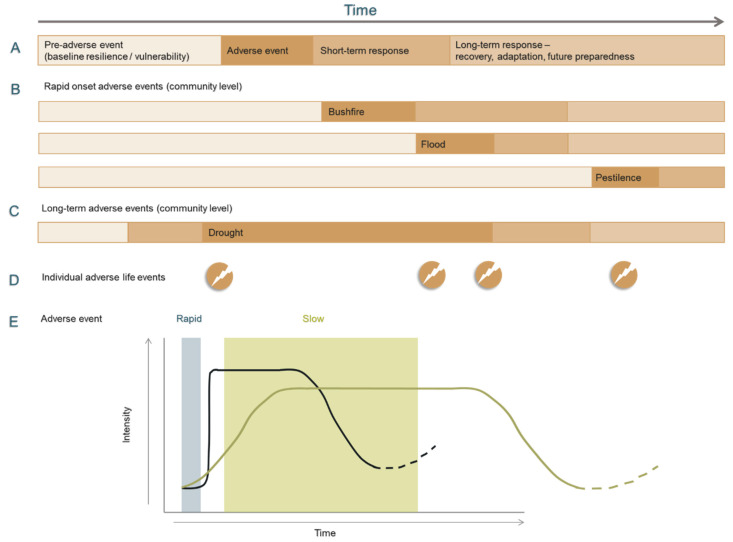
Depiction of how adverse events vary with onset and duration over time. (**A**) Schematic of phases associated with adverse events, with the pre-adverse event phase associated with baseline vulnerabilities and resilience potential. (**B**) Community-level rapid onset adverse events, examples include bushfire, flood and pestilence. (**C**) long term adverse events such as drought, where the phase edges are less clear—when drought initiates, when it ceases, with a long recovery and adaptation phase. (**D**) Individual adverse life events, examples include bereavement, serious illness, financial hardship, relationship breakdown etc. The intensity of impact is mapped for adverse events against time, with a comparison example of rapid and slow onset (**E**).

**Figure 3 ijerph-17-07205-f003:**
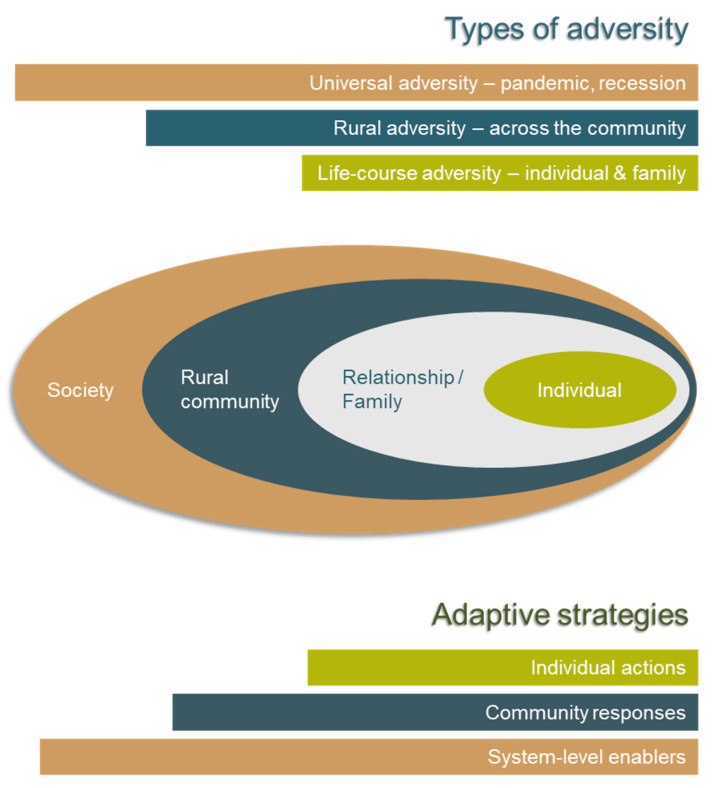
An ecological view of the impacts of adverse events at the individual (micro), community (meso) and wider system (macro) levels, and the adaptive strategies that influence preparedness, response and recovery. Ecological model adapted from [52] and tailored for rural adversity.

**Figure 4 ijerph-17-07205-f004:**
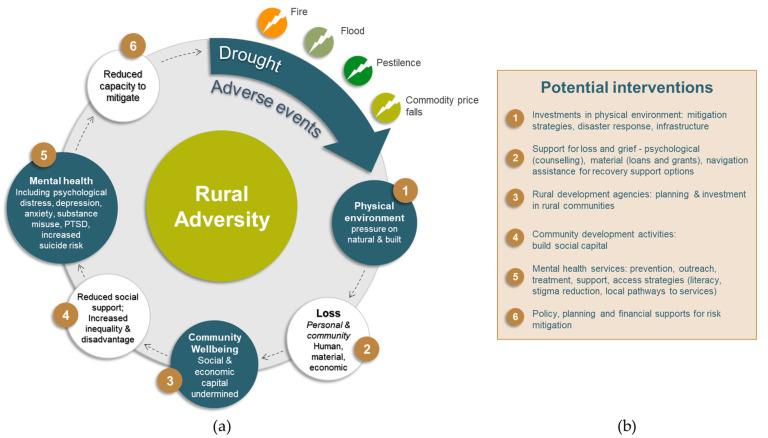
A dynamic conceptual model of rural adversity shows the impact of adverse events on the environment, community wellbeing and individual mental health (**a**) revised and adapted from [16]. Potential interventions to support mental health and mitigate against adverse events and their mapped impacts (**b**) (see main text for detailed description).

**Table 1 ijerph-17-07205-t001:** Rethinking rural adversity—summary of key workshop themes.

Workshop Themes	Rethinking and Reframing Rural Adversity
Rural resident centred	Moving from fragmented and segmented views to a holistic approach with rural people at the centre Considering how policy for an average person in an average place can be translated into local, contextually relevant responses for real people that recognises individual diversity within complex multi-layered social contexts
Rural diversity	A contextual view of rural variation rather than simple urban/rural comparisons; understanding place as a key variable
Rural adversity	Moving beyond discrete disaster responses to the inclusion of spatial, scaled, temporal, cumulative, and contextual impacts
Ecosystems approach to mental health	Adopting robust and sustainable complex adaptive systems rather than focussing on the efficiency of individual sub-systems Recognising the value of informal supports and support networks at individual, family, community and systems levels
Data and methods	Recognising the limitations of current information and evidence and adopting new ways of visualising data in contextually relevant forms Using new insights and methods to guide implementation processes—using evidence to support decision-making Empowering rural people to lead and guide responses to rural adversity

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
