# Peer review of "What Is Rural Adversity, How Does It Affect Wellbeing and What Are the Implications for Action?"

_ijerph, 2020, doi:10.3390/ijerph17197205_

Round 1

Reviewer 1 Report

Thank you for the opportunity to review the manuscript entitled "What is rural adversity, how does it affect wellbeing and what are the implications for action?"

I enjoyed reading this manuscript and believe it may be fit for publication in IJERPH. However, I do have some minor questions that I feel ought to be answered prior to potential publication. 

  1. Please perform an additional spell check on the manuscript. There were some instances of spelling mistakes (eg l.111 well-bieng; l.125 and and). 
  2. The figures provided a lot of additional insight. The visualisation of the models definitely helps the readers grasp the content. However, please also include sources for these models. 
  3. As to Figure 3, I was wondering on what basis the model was proposed. I understand that this was adapted from Hart et al 2011 as a result of the literature review and the discussion session, but please make this more concrete. 
  4. The current paper is a concept paper, based on literature review and expert input. However, it would have been good to have some quantitative data in relation to the model proposed in figure 3. This was, of course, not the set-up of the current paper, but should, at the very least, be mentioned in more detail in the recommendations section of the discussion.

I wish the authors well in revising their interesting manuscript. 

Author Response

We thank the reviewer for their helpful comments - please see the attachment for our responses.

Reviewer 2 Report

A theoretical and conceptual paper on rural adversity in relation to mental health and wellbeing is a welcome contribution to the field. In particular, my interest was piqued by the suggestion that the paper would incorporate theoretical insights from ‘other disciplines’ (presumably other to Health) in relation to ‘place’ and the complex interplay of social, economic and cultural factors that shape rural places.  It is suggested that a turn to geography for such insights would be a useful exercise – to which I would agree having delved myself into geographies of rural mental health. However, to my disappointment it seems that this direction was not pursued in the paper after all and there was no mention of geography – human, social or cultural - in the subsequent methods section where the multiple disciplines consulted for the review of literature were listed or the disciplines of the expert group identified.  

I then wondered whether the stated aim to review the Hart et al model ‘from different theoretical perspectives’ may suggest a scope that is too much to chew in a single paper. In the introduction there was indeed much going on – with reference to the Hart model, the mental health ecosystems approach, the six factor framework, complex social justice issues and the suggestion for structural reform to the service system. However, by the end of the introduction I was still unsure about the perceived strengths and limitations of the Hart model and why the contribution of the paper to adapt or extend this model in the way the authors proposed was warranted.

My recommendation to the authors would be to sharpen and clarify the argument in the introduction by identifying the theoretical or conceptual ‘problem’ they have identified and clearly outline their approach to the problem and the contribution of the manuscript. And if the various fields of geography do not have a substantial role to play then perhaps refine where the ‘different theoretical perspectives’ are derived from and why.

I found the empirical approach to developing theory interesting– It’s not common practice in the social sciences. However, given that an empirical approach was taken, I found that the paper fell short in terms of outlining the ‘results’ – what contribution did the expert group make to the findings? How were disagreements resolved? What were the key outcomes that informed the current manuscript? Some overall presentation of key insights, themes or other data would be useful at this stage before diving into the detailed discussion of each section. 

I found section 3.1 after ‘Results’ disorienting – From the introduction I was expecting that a theoretical articulation of place would occur at some point – but anticipated that it would be situated in relation to the Hart model in some way (which at this stage I am still unclear about in terms of detail about the model and its perceived shortcomings). As I read on I found this section very descriptive of well understood themes from the literature and was disappointed it did not offer a deeper articulation of ‘place’ in relation to adversity – this is captured at the end of the paragraph with ‘We can conclude that rural communities are both diverse and dynamic’.

The following section follows a similar pattern with descriptive commentary and little connection back to what these insights mean for the model. I wasn’t able to follow a clear argument or theoretical development – which I think is overall what this paper is missing. I would recommend some restructuring of the paper to more clearly articulate the authors argument around what the new literature/understandings means for the model and theoretical conceptions of rural adversity and providing a narrative through the paper that clearly demonstrates what is being advanced in each section. This would strengthen the contribution of the paper to the literature and therefore potentially its impact on the field.

In section 3.2 the authors have an opportunity to consult geography to discuss how a ‘multi-scalar’ approach provides insight into the mental health of individuals in relation to subjectivity, family, community….out to global or wider social constructs. This might help nuance this discussion further.      

In section 3.4 we finally get to the new model – I think it needed to be presented earlier with some narrative about how the expert group provided input into its development and then ‘unpacked’ in subsequent sections. If this is the core contribution of the paper, I think it needs to be up front and centre stage. I note that the model is still quite sociological in its orientation despite mentions of ‘cultural factors’ earlier in the piece. If the model does incorporate cultural dimensions of rural adversity – such as rural subjectivities and cultural practices that can impact on mental health and help-seeking then perhaps these need to be flagged – or perhaps don’t include cultural earlier in the paper.

I am pretty sure the authors have identified a worthwhile contribution to understandings of rural adversity within the discipline of health – and I laud their efforts to draw on disciplines from outside health to do so. That is not an easy feat.  So I think this paper hold promise – and just requires some further development and refinement to clearly communicate their important message and really drive home their recommendations for intervention.       

Author Response

We thank reviewer 2 for challenging and very helpful comments, they have helped us deliver a much better paper. Our responses are addressed in the attached document.

Reviewer 3 Report

The authors have performed a literature review and discussed with an expert disciplinary group to update the rural adversity model put forward by Hart et al. They also discuss the implications of their updated model.

The aim of this study is important because arguably, the increasing adversity due to climate change and the current pandemic is disproportionally affecting people living in rural areas. My expertise is not in climate science, however, so I will focus my review on the mental health elements of the study. The manuscript is well structured and clearly written. Below I list some comments that I hope will help the authors improve their manuscript further.

  1. I would appreciate a more detailed description of the literature review, including dates and specific search strings, preferably with a flow chart of the selection process. This could be put in a supplement, but it is pretty relevant information to understand the quality of the literature review.

Compared to the introduction, I feel that the results section could use more work. For example:

  1. The authors write in section 3.1: “Contextualising theories and concepts of ‘intersectionality’ can be useful for relational understanding of rural variability that can result in different levels of disadvantage for different rural communities.” Yet they don’t really mention any specific theories or concepts in their work after this. I would either remove this or go more in-depth. It’s also mentioned in the discussion, where it also falls a bit flat for the same reason. Also, the sections about topophilia and solastalgia seem a little bit out of place in section 3.1, and the order of discussed topics does not make the most sense to me. I would have chosen something more like:
  • General challenges of living in remote areas
  • Population decline and it’s (spiralling) effects
  • The challenges of providing mental health services in rural areas

Finally, the authors conclude: “We can conclude that rural communities are both diverse and dynamic hence the importance of using an ecosystems approach to conceptualise rural adversity more comprehensively.”, yet they do not really emphasize these things in their description of the rural context.

  1. In section 3.2, I would start with the definition of adversity, and the explanation that it can be on different levels (i.e. personal, community, nation, world). It makes more sense to me to introduce the concept of rural adversity after, coupled with the key aspects from Hart’s model, which are now at the end. The discussion of the different levels of adversity and how they interact could be more streamlined in the last paragraph. This sentence is in there twice: “Global adversity of which the COVID-19 pandemic and global economic recession are clear examples which may also overshadow and exacerbate community-level adversities.”
  2. In the end of section 3.3., the authors write: “Developing the concept of rural adversity with the evidence presented here is by no means 348 exhaustive. […] Moreover a targeted focus on the poor mental wellness of children and adolescents in 352 rural Australia can help understanding of longer-term outcomes of adversity for rural 353 communities [88].” I would move this section to the discussion.
  3. The authors often mention the pandemic as an example, but do not elaborate on the exact impacts of the pandemic and the subsequent recession.

Most of the figures could use some tweaks/more explanation.

  1. I’m a little bit unclear about figure 1. Is this a theoretical model to show how different types of adversity have different time frames? Individual adverse events also have the characteristics described in section A. It’s not clear from the section titles that B and C are about community events, but they must be, based on the examples. All in all, figure 1 looks more like what the life of a single person in rural Australia might be like, but that’s not what the caption says. There does not seem to be any theoretical background to the specific examples (i.e. the short-term response of a bush-fire is twice as long as that of a flood), so perhaps the authors could emphasize somewhere this is just an example of what these effects could look like in a specific person, but might differ between people/communities
  2. Section 3.3 is very clear, but the text in figure 2 could be improved. After ‘universal adversity’ it gives examples, but after ‘rural adversity’ and ‘life-course’ it gives an explanation. It would be better if they are all examples or all explanations.
  3. Figure 3 also contains some unclear text. It says the boxes are potential interventions. But then at the “loss” bubble, the box says: “Loss & Grief”. How is that an intervention? And near the bubble with “Physical environment” it says “Prevention”, but prevention of what? The adverse event has already happened at that point. Mitigation might be more appropriate here. Other than that, the model seems sensible.

Author Response

We thank Reviewer 3 for their thoughtful and helpful feedback, our responses are in the attached document.

Round 2

Reviewer 2 Report

I agree that the revisions have significantly improved the manuscript and I am pleased I was able to provide useful comments in that regard. 

The introduction is far sharper providing greater clarity on the contributions of the paper. The additional paragraphs and insertions have also clarified the argument and its conceptual basis and significantly strengthened the piece. The methods section is excellent. 

The paper now reads beautifully and I found it lively and rich. 

There were just a few minor editing issues:

line 56 problematic rather than problematical

line 58 a better word than 'fixes'  

line 190 review sentence re movement - as both versions are struck out

line 286 'Universal adversity, of which the COVID-19 pandemic and global economic recession are clear examples, may also overshadow' 

line 334 delete 'that' 

line 504 delete 'the fact' 

line 553 I think the authors could rightfully 'anticipate' rather than 'hope' that the paper will resonate internationally.  

Author Response

We thank reviewer 2 for their time and effort in helping us improve this paper. In response to the second review, we acknowledge the following corrections in line with their suggestions. All revisions have been done in 'track changes'.

line 56 problematic rather than problematical - changed to problematic (now p2 line 48)

line 58 a better word than 'fixes' – substituted ‘reparation’ (p2 line 48).

line 190 review sentence re movement - as both versions are struck out – Sentence now reads ‘These losses may cause population movement to places of higher amenity’ (p5 line 157)

line 286 'Universal adversity, of which the COVID-19 pandemic and global economic recession are clear examples, may also overshadow' – this has been changed to ‘Universal adversity, of which the COVID-19 pandemic and global economic recession are clear examples, may overshadow and exacerbate community-level adversities.’ (p7 line 227)

line 334 delete 'that' – done p8 line 259

line 504 delete 'the fact' – done p10 line 336

line 553 I think the authors could rightfully 'anticipate' rather than 'hope' that the paper will resonate internationally.  – done p10 line 375.